# Sample Environment for Operando Hard X-ray Tomography—An Enabling Technology for Multimodal Characterization in Heterogeneous Catalysis

Johannes Becher [1], Sebastian Weber [1,2], Dario Ferreira Sanchez [3], Dmitry E. Doronkin [1,2], Jan Garrevoet [4], Gerald Falkenberg [4], Debora Motta Meira [5,†,‡], Sakura Pascarelli [5,§], Jan-Dierk Grunwaldt [1,2] and Thomas L. Sheppard [1,2,*]

1 Institute for Chemical Technology and Polymer Chemistry (ITCP), Karlsruhe Institute of Technology (KIT), Engesserstr. 20, 76131 Karlsruhe, Germany; johannes.becher@kit.edu (J.B.); sebastian.weber@kit.edu (S.W.); dmitry.doronkin@kit.edu (D.E.D.); grunwaldt@kit.edu (J.-D.G.)
2 Institute of Catalysis Research and Technology (IKFT), Karlsruhe Institute of Technology (KIT), Hermann-von-Helmholtz Platz 1, 76344 Eggenstein-Leopoldshafen, Germany
3 Paul Scherrer Institut (PSI), 5252 Villigen, Switzerland; Dario.Ferreira@psi.ch
4 Deutsches Elektronen-Synchrotron DESY, Notkestrasse 85, 22607 Hamburg, Germany; jan.garrevoet@desy.de (J.G.); gerald.falkenberg@desy.de (G.F.)
5 European Synchrotron Radiation Facility (ESRF), 71 Avenue des Martyrs, 38000 Grenoble, France; dmeira@anl.gov (D.M.M.); sakura.pascarelli@xfel.eu (S.P.)
* Correspondence: thomas.sheppard@kit.edu; Tel.: +49-721-608-47989
† Current affiliation: CLS@APS Sector 20, Advanced Photon Source, Argonne National Laboratory, 9700 S. Cass Avenue, Argonne, IL 60439, USA.
‡ Current affiliation: Canadian Light Source Inc., 44 Innovation Boulevard, Saskatoon, SK S7N 2V3, Canada.
§ Current affiliation: European XFEL GmbH, Holzkoppel 4, 22869 Schenefeld, Germany.

**Abstract:** Structure–activity relations in heterogeneous catalysis can be revealed through in situ and operando measurements of catalysts in their active state. While hard X-ray tomography is an ideal method for non-invasive, multimodal 3D structural characterization on the micron to nm scale, performing tomography under controlled gas and temperature conditions is challenging. Here, we present a flexible sample environment for operando hard X-ray tomography at synchrotron radiation sources. The setup features are discussed, with demonstrations of operando powder X-ray diffraction tomography (XRD-CT) and energy-dispersive tomographic X-ray absorption spectroscopy (ED-XAS-CT). Catalysts for $CO_2$ methanation and partial oxidation of methane are shown as case studies. The setup can be adapted for different hard X-ray microscopy, spectroscopy, or scattering synchrotron radiation beamlines, is compatible with absorption, diffraction, fluorescence, and phase-contrast imaging, and can operate with scanning focused beam or full-field acquisition mode. We present an accessible methodology for operando hard X-ray tomography studies, which offer a unique source of 3D spatially resolved characterization data unavailable to contemporary methods.

**Keywords:** in situ; operando; synchrotron radiation; X-ray tomography

## 1. Introduction

Derivation of structure–activity relations is a major goal of modern heterogeneous catalysis research. These relations link physical sample composition (e.g., distribution of active metal, pore network characteristics, shape and structure of support) with catalytic performance (e.g., reactant conversion, product selectivity, reaction mechanism) [1,2]. Understanding structure–activity relations is a challenging yet crucial step in developing more efficient and stable catalysts by promoting knowledge-based design and operation of catalytic processes, rather than trial and error experimental or synthetic approaches [3,4]. Over the last decade or more, in situ and operando characterization have been established

as essential concepts for deriving such structure–activity relations in catalysis [5–7]. However, while in situ and operando studies are now commonplace, often these are coupled to averaging measurements, which only address global sample composition (e.g., powder X-ray diffraction (XRD), diffuse reflectance infrared spectroscopy (DRIFTS), or X-ray absorption spectroscopy (XAS)). This can be problematic since samples in heterogeneous catalysis are typically rather complex or hierarchically structured. These may consist of composite materials or contain diverse macroscopic features, internal pore networks on the μm to nm scale, together with active catalytic sites (e.g., metal nanoparticles, clusters), and various support or spectator phases [8–10]. Due to this intrinsic heterogeneity, comprehensive in situ and operando analysis of a target catalyst or process should ideally cover all relevant length scales [11–13] and ideally include both global and spatially resolved measurements to produce meaningful results regarding local sample structure [14,15]. The dynamic nature of catalytic processes should also be addressed where possible, including structural or chemical gradients, which may be present within catalyst particles, beds, and reactors [16–18]. In summary, effective catalyst characterization clearly requires a careful approach. The more accurate and realistic the characterization in spatial and time domains and in terms of reaction conditions, the more information can be harvested regarding structure–activity relationships.

The development of operando methodology has particular synergy with synchrotron light sources, which offer a diverse toolkit of hard X-ray techniques for catalyst characterization. XAS and XRD are just two of many widely used examples of global characterization tools in catalysis research [19–21]. Hard X-rays can also be focused on micro- and nanobeams (pencil beam mode) or applied in parallel beam geometry with 2D area detectors (full-field mode). This enables a range of spatially resolved characterization tools collectively known as X-ray microscopy [22–24]. By further applying the principle of tomography, it is possible to obtain 3D chemical images of an interior and exterior sample structure in a non-invasive manner, with spatial resolution on the μm or nm scale. A range of X-ray contrast modes can furthermore be combined with tomography, producing 3D spatially resolved data based on XAS, XRD, or X-ray fluorescence (XRF); for example, The diverse contrast modes and applicability to extended sample volumes present a broad range of possibilities for catalyst characterization. For example, hard X-ray tomography has previously been applied ex situ to quantify porous features and map deactivation processes such as metal sintering or poisoning [25–31]. In particular, tomography is gaining attention among the catalysis community for delivering structural insights without invasive sample preparation, which are often unavailable using global or even 2D spatially resolved methods [32,33]. However, the majority of studies in the literature to date have been performed ex situ. The feasibility of operando hard X-ray tomography studies in addressing catalyst structure–activity relations is heavily dependent on the availability of appropriate sample environments [34,35]. For operando hard X-ray tomography, experimental complexity is a considerable challenge, requiring careful sample preparation, precise positioning of the reactor (μm or nm scale), free translation and rotation of the sample within the beam, application of realistic process conditions (e.g., temperature, gas environment), and some means of product detection. For this reason, in situ 3D imaging of catalysts at work has only been marginally explored [36–39], with even fewer operando studies known in the literature [40–42]. The optimum case of combining 3D structural observations with quantitative catalytic performance data, which is necessary for an accurate discussion of structure–activity relations, was only recently reported [43]. Coupling operando methodology to various forms of X-ray tomography, therefore, constitutes a significant unexploited potential in catalysis and materials research, necessitating the development of dedicated sample environments to perform such studies.

Here we present a versatile sample environment optimized for operando hard X-ray tomography of catalysts and functional materials at work—the rotating capillary for tomographic in situ/operando catalysis (aRCTIC). The aRCTIC setup was designed for beamline P06 at the PETRA III synchrotron radiation source. The design of aRCTIC and its

suitability for catalysis and materials research are presented, emphasizing the flexibility of the measurement concept, which is easily adaptable to other beamlines. As a first case study, operando XRD tomography is demonstrated in a packed bed reactor containing a $Ni/Al_2O_3$ catalyst used for the $CO_2$ methanation reaction. 3D spatially resolved XRD patterns were recorded with ca. 20 μm spatial resolution under reaction conditions, with simultaneous quantitative reactant and product analysis by online mass spectrometry (MS). A second case study shows catalytic partial oxidation of methane over a $Pt/Al_2O_3$ catalyst while outlining the feasibility of operando energy-dispersive (ED-XAS) tomography under reaction conditions. The aRCTIC setup is presented as an enabling technology aimed at generally improving access to various forms of operando hard X-ray tomography, particularly within the catalysis and energy materials communities.

## 2. Results and Discussion

### 2.1. Design of the aRCTIC Setup

The aRCTIC setup was designed for use at hard X-ray microscopy beamlines operating with either focused scanning beam or full-field imaging modes but may also be adapted for spectroscopy and scattering beamlines. The setup allows studies on model powder catalysts, structured catalyst particles, packed-beds, or mm-scale catalytic reactors under controlled temperature and gas conditions. An illustration of aRCTIC installed at beamline P06 of PETRA III is shown in Figure 1 (see Supplementary Materials for further images). The aRCTIC setup is based on the following design principles:

- A stable sample holder and support structure, allowing μm precise translation and maintaining the sample within the center of rotation;
- Free rotation in a complete 180° arc, avoiding missing wedge artifacts from incomplete tomography scans or incomplete sinograms;
- A support rod relatively transparent to hard X-rays and with minimal scattering, minimizing image artifacts at angles where the support rod may eclipse the X-ray beam;
- A closed gas-tight system allowing precise control of gas composition, flow rate and space velocity;
- An integrated online product analysis for operando measurement and quantification of catalyst performance, in terms of gas-phase reactant and product composition;
- A uniform source of heating to maintain stable catalytic performance during rotation and translation motions;
- A compact design for integration in beamline stages with a limited weight allowance. Small profile to allow positioning of optical components, detectors, other beamline apparatus, etc.

The aRCTIC setup is modular by design, whereby many of the components can be interchanged or substituted with components other than those indicated here. The aRCTIC concept, therefore, supports integration at a variety of hard X-ray beamlines, which may have different infrastructure and arrangements of devices. Examples of this modularity are indicated throughout the following sections.

To meet the requirements for operando catalysis studies, particularly the need for a gas-tight system and controlled environment, a quartz capillary reactor was chosen as the primary sample holder. Such capillary reactors are analogous to those used in many conventional XAS or XRD studies using synchrotron radiation [44]. A tube or rod (support) made of glassy carbon is positioned parallel to the capillary sample holder. The support provides rigidity, bears the weight of the stainless-steel gas connections at the top of the setup, and absorbs torsional force during rotation of the sample holder. This is critical due to the fragility of quartz capillaries. To perform tomography, the sample holder is placed on top of a motor system, normally consisting of a pair of linear stages (for center of rotation alignment), a rotation stage, and optionally a goniometer for vertical alignment (see the "experimental" section for further information). One issue with aRCTIC is that the sample and support must be rotated through a 180° degrees for optimum tomography measurements. This means the support will inevitably eclipse the incident X-ray beam

across a small angular range (depending on support diameter, e.g., 5–10° at 1–3 mm diameter). Glassy carbon was, therefore, chosen as the support material due to its favorable mechanical strength, low thermal expansion coefficient, and low attenuation coefficient in the hard X-ray regime. Depending on the incident beam energy and the detection method used, the presence of the support in the measured data can either be ignored or corrected during tomography reconstruction (see Supplementary Materials). With the aid of support, gas-tight connections can be maintained at the inlet and outlet of the sample holder.

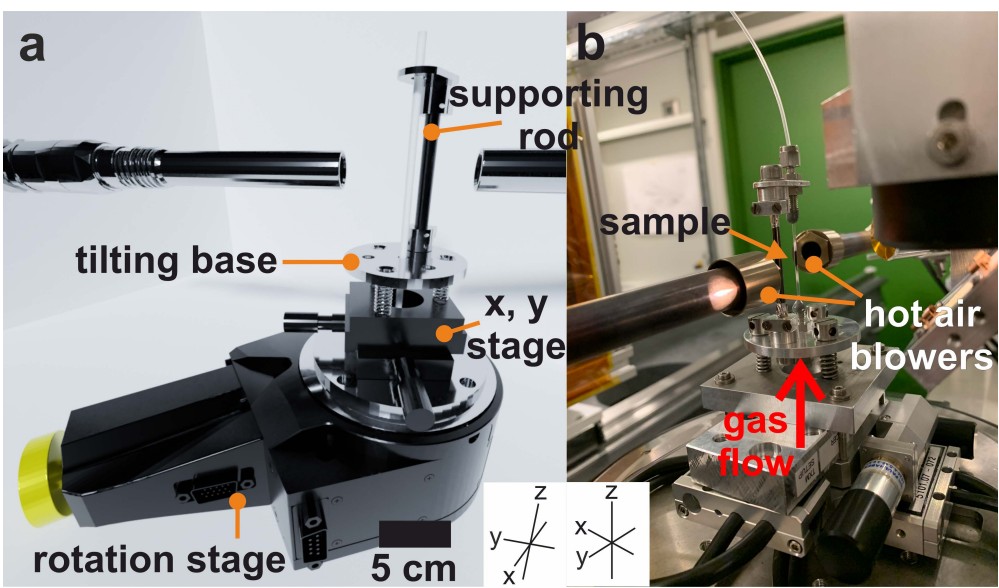

**Figure 1.** (**a**) 3D rendering of the rotating capillary for tomographic in situ/operando catalysis (aRCTIC) setup; (**b**) aRCTIC installed at the microprobe end station of beamline P06 at PETRA III—showing 1 mm quartz capillary mounted with support rod, twin hot air blowers and gas-tight fittings.

An appropriate gas delivery system, such as thermal mass flow controllers, allows precise control of gas composition, flow rate, and space velocity analogously to conventional XAS or XRD studies in capillaries. Two hot air blowers positioned on either side of the sample were chosen to allow relatively uniform heating, analogous to those used in conventional XAS and XRD studies [44]. Particular care must be taken when using aRCTIC to ensure uniform heating is applied to the sample during rotation and for extended periods of tomography data acquisition. Alternative heat sources, such as resistive heating coils, halogen lamps, or infra-red heaters, are feasible alternatives, provided that they do not eclipse the incoming or outgoing X-ray beam or deliver unwanted heat to nearby beamline instrumentation (e.g., detectors, optics). A gas-phase product analysis method, such as MS (shown in this work) or gas chromatography (GC), can readily be attached downstream of the reactor to sample and quantify the composition of the gas stream. Crucially, this can also be used to ensure that the system remains gas-tight and to monitor abnormalities, resulting from the movement of the sample out of the heated zone. This completes the essential elements for operando tomography studies of gas-phase catalytic processes: (i) aRCTIC sample holder (capillary, support, fittings); (ii) gas control system; (iii) heat control system; (iv) product analysis system.

Among the components listed above, the motor system used for tomography acquisition is generally unique to each individual beamline and is, therefore, not defined as a component of the aRCTIC setup directly. The setup can be flexibly applied at other beamlines, provided that a rotation stage, linear centering stages, and optionally a goniometer for vertical alignment of the capillary are present. In the implementation of aRCTIC described here, the goniometer and linear centering stages were both manually

operated. This allowed sample positioning with μm-scale precision, sufficient for many microtomography or full-field imaging experiments.

### 2.2. Operando XRD-CT Case Study—Catalytic Performance during $CO_2$ Methanation

A $Ni/Al_2O_3$ catalyst previously reported in the literature was studied in the form of a dry gel received directly after synthesis [45]. The catalyst is active in $CO_2$ methanation to $CH_4$, according to Equation (1):

$$CO_2 + 4H_2 \rightleftharpoons CH_4 + 2H_2O \qquad (1)$$

At low temperatures, the forward reaction is predominant. The catalyst was investigated under five experimental conditions: (i) initial state—dry gel measured ex situ; (ii) activation in reducing conditions (823 K, 25/75% $H_2$/He, 10 mL/min); (iii) reaction conditions (673 K, 20/5/75% $H_2$/$CO_2$/He, 10 mL/min); (iv) thermal aging—reaction conditions at 973 K; (v) reaction conditions at 673 K after thermal aging. XRD tomography measurements were performed at stages (i), (ii), (iii) and (v). 2D XRD line scans were continuously performed when changing conditions to ensure the catalyst was stable before beginning each of the longer tomography scans. Catalytic performance data during the different stages of the operando study starting from activation (ii) are depicted in Figure 2.

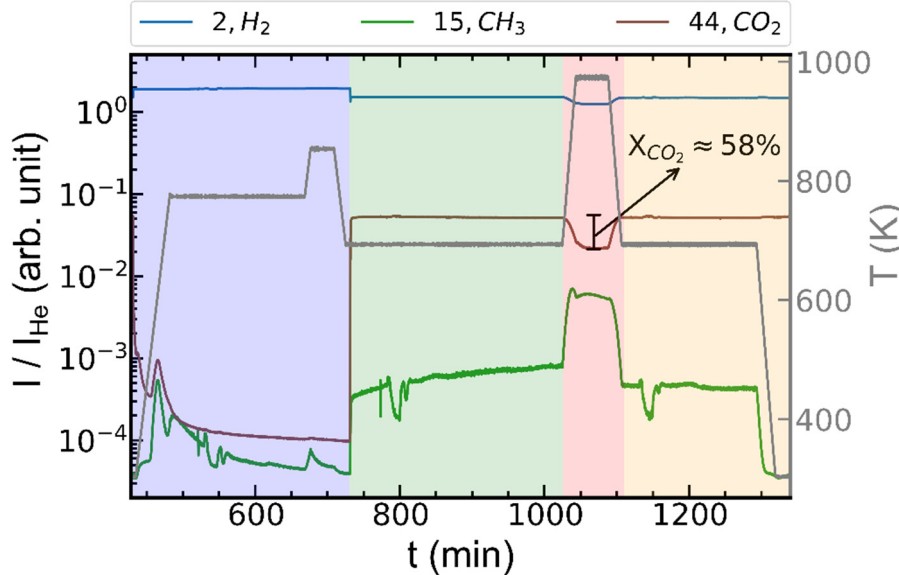

**Figure 2.** Mass spectrometry (MS) data showing consumption of $H_2$ (m/z = 2) and $CO_2$ (m/z = 44), and formation of $CH_4$ (attributed to $CH_3^+$ ion m/z = 15) during tomography experiments. Traces were normalized to He (m/z = 4). Colored regions indicate: activation (blue); reaction conditions (green); thermal aging (red); reaction conditions after aging (orange). $X_{CO2}$ indicates $CO_2$ conversion during thermal aging. Timescale includes 3 tomography measurement series and condition changes.

The MS data are shown in Figure 2 allow quantification of $CO_2$ conversion (see Supplementary Materials for calculations). All traces were normalized to the He trace at m/z = 4 as an internal standard, the flow of which remained constant throughout the experiments (see Supplementary Materials for raw data). The temperatures shown are the experimentally measured and calibrated values, not the set point of the heating system (see Supplementary Materials for calibration data). From the MS data $CO_2^+$ (m/z = 44) and $CH_3^+$ (m/z = 15) fragments were detected in the initial stages of activation (up to ca. 500 min, Figure 2—blue). These originate from the decomposition of organic compounds used as porogenic agents (e.g., dodecylamine), which are still present in the dry gel after synthesis [45]. Switching from activation conditions to reaction conditions (ca. 730 min, Figure 2—green), one can observe the direct increase in the $CH_3^+$ fragment, indicative of

CH$_4$ formation. The catalyst activity increased gradually during the reaction condition-stage (ca. 730 to 1030 min), and steady-state conversion was not reached. Tomography was performed in any case due to the relatively small change in CH$_3^+$ signal during this time (Figure 2, note logarithmic scale). High-temperature aging of the catalyst (ca. 1030 to 1110 min, Figure 2—red) temporarily led to higher conversion of H$_2$ and CO$_2$ (up to 58% conversion) and increased yield of CH$_4$. Switching back to reaction conditions (ca. 1110 min, Figure 2—orange), one can observe a decrease in CH$_4$ formation compared to the first stage of reaction conditions (Figure 2—green). Finally, upon cooling to 323 K, CH$_4$ was not observed anymore. This illustrates accurate monitoring, control, and quantification of the composition of the outlet gas stream during tomography studies in the aRCTIC setup. The gas-tight nature of the system can furthermore be proven by monitoring the Ar$^+$ trace (m/z = 40), which remained at negligible levels during tomography studies. Ar is present in small amounts in ambient air (9340 ppm) and can, therefore, indicate the presence of leaks (see Supplementary Materials). Small fluctuations in methane signal observed starting at around 770 min, and 1140 min resulted from the translate-rotate measurement scheme employed. The supporting rod partly covered the exit of one hot air blower over a small angular range. This issue can be avoided by implementing a rotate-translate measurement scheme to minimize the obstruction of the hot air blower or by introducing a slight angle between the two hot air blowers so that the support does not obstruct them at any time.

### 2.3. Operando XRD-CT Case Study—Tomography Measurements

Following tomography data acquisition and reconstruction, a series of cross-sections (slices) of the catalyst bed was acquired over each measured condition, excluding during thermal aging. Each pixel or sampling point within each slice contains a full XRD pattern, which can be resolved independently. An example of scanned slices at each measurement condition is shown in Figure 3. These images represent the total signal acquired across all angles in the 2θ range of 0 to 22° during XRD tomography. Note that the slices shown do not represent the same physical position in the catalyst bed. Due to the absence of locative markers and the small beam size in this study, a small variation in the scanned position was tolerated. Measurement at identical positions can be performed according to the needs of the experiment by implementing markers, for example.

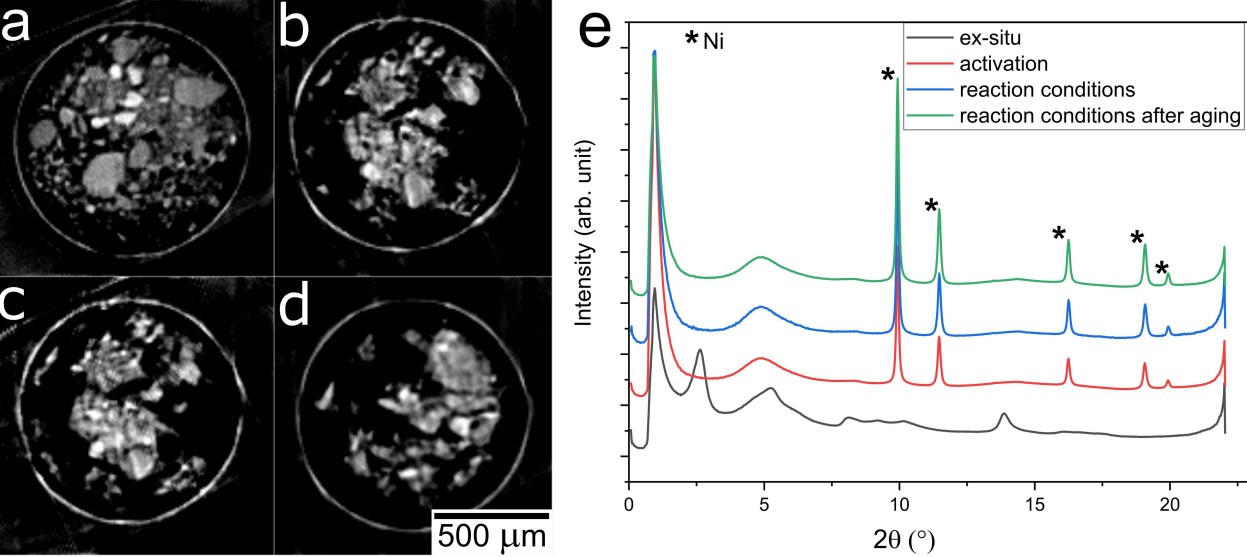

**Figure 3.** Summed intensities of all reflections for slices at the following conditions: (**a**) initial state; (**b**) activation; (**c**) reaction conditions; (**d**) reaction conditions after thermal aging. (**e**) Normalized sum XRD patterns of the whole measured slices of the Ni/Al$_2$O$_3$ catalyst, representing total signal from all pixels of the XRD tomography images. XRD patterns recorded at λ = 0.0354 nm.

The particles visible in the reconstructions can be analyzed regarding the crystalline phase composition or crystallite sizes in the same way as for normal XRD patterns. However, in principle, XRD tomography allows such analysis to be performed on a pixel-by-pixel basis. The tomograms are, therefore, more sensitive to heterogeneity in the sample of interest compared to standard powder XRD measurements [40,41]. Here the pixel size of the reconstructed data was around 10 μm with an estimated resolution of about 20 μm (see Supplementary Materials). The sum XRD patterns of the four measured slices are shown in Figure 3e. These patterns can be considered analogous to a conventional powder XRD measurement in that they represent the sum signal from all pixels in the images from Figure 3. The diffractograms measured under reaction conditions clearly differ significantly from the original ex situ measurement of the dry gel. Prior to activation, the $Ni/Al_2O_3$ catalyst shows reflections probably originating from an Al-oxide-hydroxide precursor phase forming the dry gel [45]. The broad feature at 5° present in all patterns is from the quartz glass capillary sample holder. Upon activation, we can observe that the precursor phase diminished, and metallic Ni reflections occurred around $2\theta$ = 10, 12, 16.5 and 19°, while no Ni or NiO phases were visible in the ex situ sample. The Ni reflections after activation show sharp profiles indicating the formation of highly crystalline Ni, compared to the relatively amorphous character of the dry gel precursor material. The absence of the initial Al-oxide-hydroxide reflections furthermore shows a change in the support structure upon activation. This is expected due to the decomposition of organic components, in agreement with the MS data (Figure 2), and the transition from dry gel to supported $Ni/Al_2O_3$ catalyst.

To demonstrate the successful acquisition of 3D spatially resolved tomography data, several particles from the initial dry gel state were selected, and the corresponding XRD patterns were plotted individually (Figure 4b). The XRD patterns were extracted from different positions of the bed to qualitatively observe differences in the crystalline structure of the sample. It should be noted that the regions of interest (ROIs) where the XRD patterns were extracted had a size of 5×5 pixels, resulting in a relatively low signal-to-noise ratio in the diffractograms compared to the sum of the entire slice, which is nevertheless sufficient for qualitative analysis. The obtained patterns are shown in Figure 4, indicating that a relatively heterogeneous distribution of nanocrystalline or partly amorphous phases was initially present in the dry gel material. The same procedure was followed for the catalyst after activation, showing variation in the intensity of Ni reflections throughout the sample (Figure 5). This indicates a heterogeneous distribution of the Ni amount following reduction (activation). Note that similar observations are challenging purely from analysis of conventional powder XRD data, analogously to the patterns shown in Figure 3e. This demonstrates the potential value of XRD tomography as a method for isolating and deconvoluting complex crystalline structures within large catalyst samples.

By switching to reaction conditions after activation, no obvious visible changes in the XRD patterns were observed across several slices. However, comparing identical reaction conditions before (iii) and after (v) the thermal aging step, one can identify an increase in the intensity of the Ni reflections, although the full width at half maximum (FWHM) of the features at 9.93 and 11.47° showed no significant sharpening of the reflections (Table 1). It should be noted that analysis of the FWHM is demonstrated on the total sum diffraction patterns due to the relatively noisy signal acquired from single pixels. This only represents the average value of the whole bed, without considering heterogeneities that can be expected from the different patterns, as shown in Figures 4b and 5. In general, analyzing the crystallite size can indicate active site sintering, which is a common deactivation phenomenon for $CO_2$ methanation catalysts and readily available from powder XRD data [46]. Although the intensity of the Ni reflections and thus the amount of crystalline Ni increased, MS analysis of the reactor outlet showed a notable decrease in the amount of methane formed between the first operando experiment and the second one following

thermal aging. This indicates that the cause of deactivation was not necessarily related to the sintering of the active Ni species in this example.

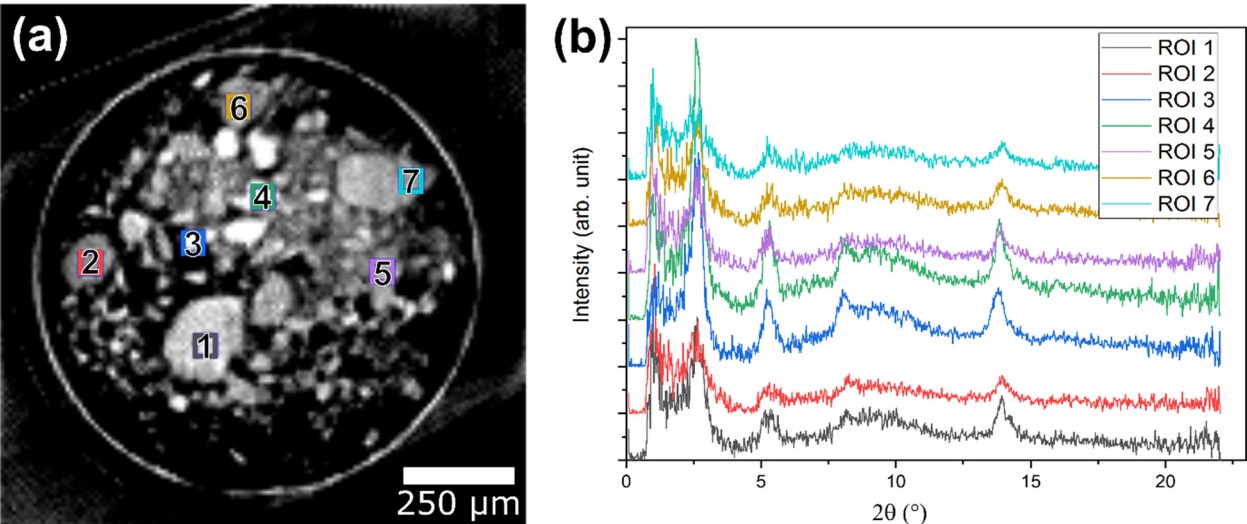

**Figure 4.** (**a**) Selected regions of interest from different particles of the catalyst bed from tomography data of the initial dry gel state. (**b**) XRD patterns represent the summed signal from all pixels within the regions of interest, indicating heterogeneity within the sample. XRD patterns recorded at λ = 0.0354 nm.

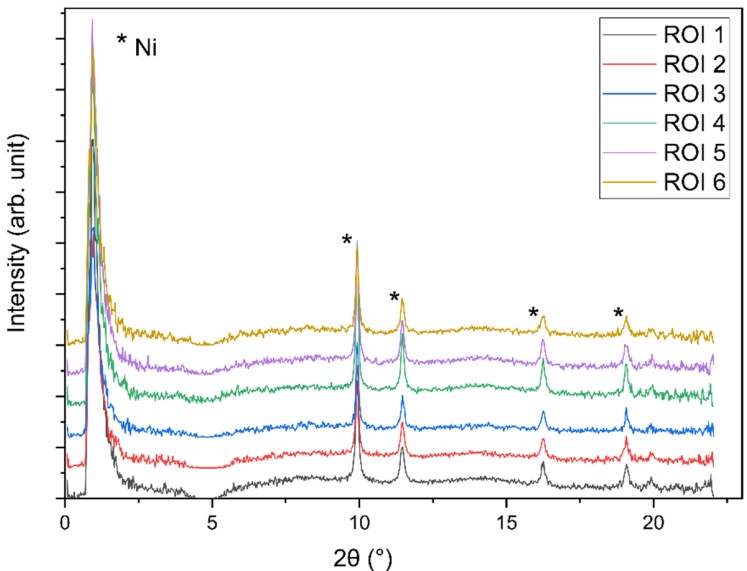

**Figure 5.** XRD patterns of selected regions of interest from tomography data of the activated catalyst after reduction. The variation in the intensity of the Ni reflections indicates heterogeneity in the crystallite size of Ni particles throughout the sample. XRD patterns recorded at λ = 0.0354 nm.

**Table 1.** Comparison of the fitted full width at half maximum (FWHM) of two Ni reflections.

| Condition | FWHM of Reflection: [1] | |
| --- | --- | --- |
| | 2θ = 9.93° | 2θ = 11.47° |
| Activation | 0.115 | 0.139 |
| Reaction conditions | 0.114 | 0.141 |
| Reaction conditions after thermal aging | 0.117 | 0.140 |

[1] Calculated error is ±0.002°.

### 2.4. ED-XAS-CT Feasibility Study—Catalytic Partial Oxidation of Methane

A feasibility study using aRCTIC was additionally performed by ED-XAS-CT on 2.2 wt % $Pt/Al_2O_3$ powder catalysts, which are active for catalytic partial oxidation of methane to synthesis gas ($CO + H_2$) [47]. The detailed synthesis procedure of this catalyst is reported elsewhere [48]. Two distinct reaction stages are possible over such $Pt/Al_2O_3$ catalysts: exothermic combustion (total oxidation) of part of available methane (2) followed by endothermic dry reforming (3) and steam reforming (4) processes. From a stoichiometric point of view, the amount of oxygen provided in the present work (2:1 $CH_4$:$O_2$) is not sufficient for the complete combustion. This allows for uncombusted methane to undergo reforming stages with combustion products resulting in the overall partial oxidation (5) as the total process.

$$CH_4 + 2O_2 \rightarrow CO_2 + 2H_2O \tag{2}$$

$$CH_4 + CO_2 \rightarrow 2CO + 2H_2 \tag{3}$$

$$CH_4 + H_2O \rightarrow CO + 3H_2 \tag{4}$$

$$CH_4 + \frac{1}{2}O_2 \rightarrow CO + 2H_2 \tag{5}$$

After an initial ex situ tomogram acquired under He flow, the catalyst was treated with 30 mL/min of 3/1.5/95.5% $CH_4/O_2/He$ at 553 K, followed by at 673 K. From the first operando measurement at 553 K, most $CH_4$ and $O_2$ was not consumed, indicating low catalytic conversion. At 673 K, $H_2$, CO and $CO_2$ were clearly observed as products together with almost complete consumption of $O_2$. A summary of MS data is shown in the Supplementary Materials. This indicates a mixture of complete combustion (2) occurring together with partial oxidation (5). Two temperature zones with different reactions (exothermic combustion and endothermic reforming) and the resulting chemical potentials ($O_2$-containing gas mixture at the inlet, $H_2$-containing gas mixture at the outlet) may, therefore, be present. These inhomogeneities suggest the formation of gradients in the platinum oxidation state, which should be potentially visible in the X-ray absorption near-edge spectra (XANES) tomography data.

### 2.5. ED-XAS-CT Feasibility Study—Tomography Measurements

ED-XAS is a developing technique allowing rapid collection of the complete XANES region using a polychromatic X-ray beam with energy and position-sensitive detector [49]. When applied to tomography, this technique is called ED-XAS-CT or tomo-XANES [50,51]. After reconstructing the tomography data, each pixel contained a full XANES spectrum and can thus be analyzed in terms of Pt oxidation state. The resolution obtained from the sharp edge fitting of the resulting tomograms was approximately 10 μm. The sum spectra obtained for the tomograms under all three measured conditions are shown in Figure 6. The pronounced whiteline feature at around 11,858 eV indicates that the as-prepared catalyst is likely fully oxidized ($Pt^{4+}$) in agreement with ex situ XAS data. This is expected since the catalyst powder was calcined in the air before use. The sequential decrease in whiteline intensity when heating to 553 K, followed by 673 K indicating a stepwise reduction from $Pt^{4+}$ towards $Pt^0$. At the highest probed temperature, the whiteline feature almost vanishes, showing a full reduction of the Pt compared to reference $Pt^0$ and $PtO_2$ spectra. These results confirm that the operando measurements and 3D tomographic XANES reconstruction were functioning as intended. In the next step, one single particle of the high-resolution operando measurement at 553 K was chosen and analyzed in detail, as shown in Figure 7. XANES spectra were extracted from different regions of interest of only $3 \times 3$ pixels resulting in a relatively low signal-to-noise ratio compared to conventional XAS. On this occasion, no visible gradient could be reliably observed, but it is rather expected that the catalyst particle shown was fully reduced. Similar results were observed for the catalyst measured at 673 K (see Supplementary Materials).

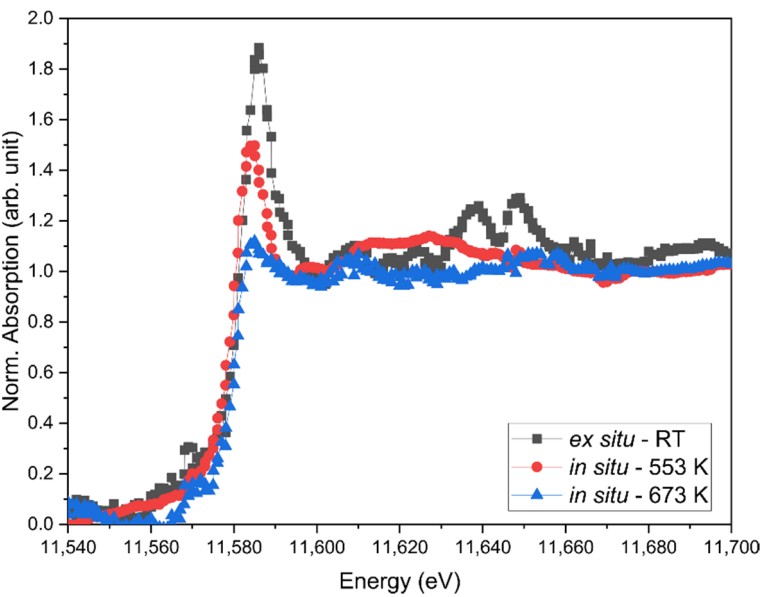

**Figure 6.** Normalised XANES sum spectra of the whole measured slices of the Pt/Al₂O₃ catalyst under the conditions indicated. This represents the total signal derived from each pixel of the ED-XAS-CT image series as each energy position.

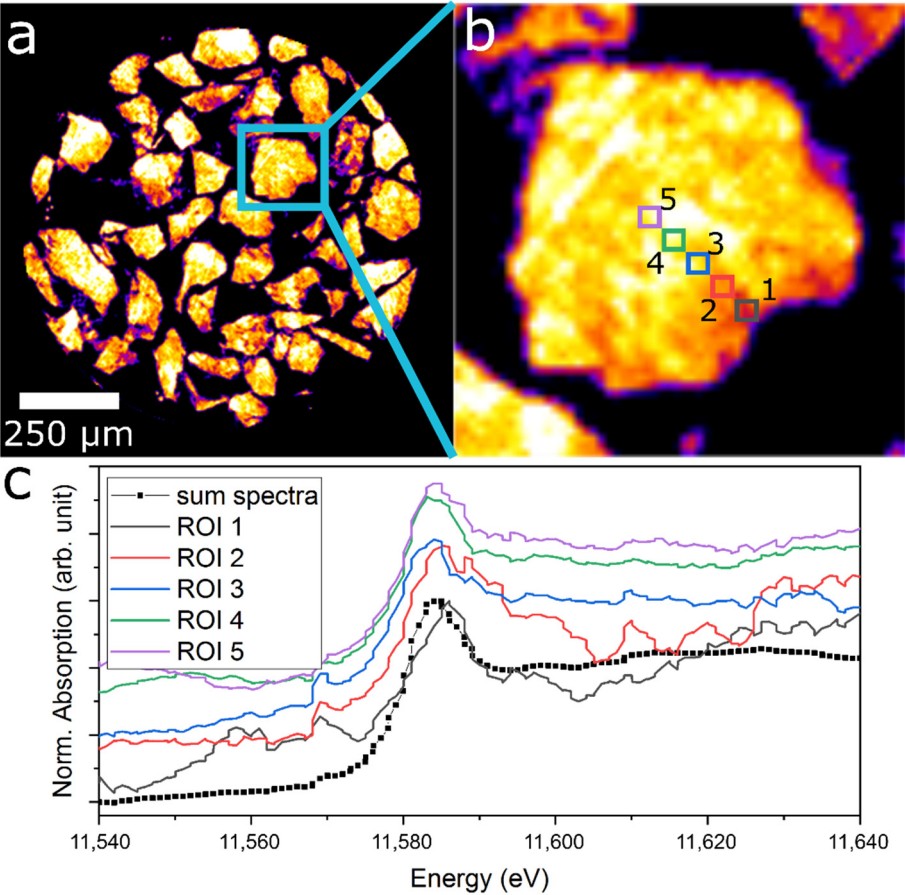

**Figure 7.** Overview of the sample measured at 553 K; (**a**) reconstructed slice with the capillary masked out and color-coding depicting the absorption intensity at 11,585 eV; (**b**) extracted single particle of the same slice with marked regions of interest; (**c**) XANES spectra extracted from each region of interest, compared with the sum spectra of the whole slice.

Although no changes in the spectra at different conditions could be found during these proof-of-concept measurements, the principle of operando tomography and ED-XAS tomography are functioning as intended. Therefore, the potential of the method is promising if improvements can be made in the signal-to-noise ratio. Although ED-XAS-CT has been successfully demonstrated under ambient conditions to probe various iron mineral phases, and iron oxidation states within a bulk catalyst particle [50,51], further investigations are needed to determine the feasibility of ED-XAS-CT for gas-phase catalysis studies. It should be noted that ED-XAS was performed here with primary slits fully open (i.e., full horizontal acceptance of the incident beam) and focused on a small $3 \times 7 \ \mu m^2$ spot, leading to relatively high flux on the sample compared to conventional XAS studies. It is possible in certain cases that beam-induced effects can likewise be more significant during ED-XAS than for conventional XAS, including desorption of adsorbed gas species [43] or unexpected variations in oxidation state induced by the beam. Nevertheless, the aRCTIC sample environment again demonstrated maintaining controlled gas and temperature environment during extended tomography studies, which is crucial for successful operando measurements.

### 2.6. Evaluation of the aRCTIC Setup and Operando Tomography

The case studies here are intended primarily to demonstrate that operando tomography measurements (in this case, XRD-CT and ED-XAS-CT) with acceptable image quality and simultaneous catalytic performance data can be obtained using the aRCTIC setup. XRD tomography, in general, can produce extensive volumes of data, consisting of individual XRD patterns for every 3D sampling point within the catalyst bed, resulting in 4D data (three spatial dimensions, 2θ and intensity). When combined with reaction conditions, such as variations in temperature, gas flow rate or gas composition, such datasets can easily stretch to more than 5D [40]. The same is true for ED-XAS-CT or more conventional XAS-CT (three spatial dimensions, energy and intensity). Strategies to comprehensively treat and extract value from these highly complex datasets are only beginning to emerge in the literature and are beyond the scope of this work [52]. It should furthermore be noted that XRD and ED-XAS tomography are only two examples of the potential studies, which can be performed using aRCTIC. Operando tomography covers a range of possible techniques, encompassing various X-ray imaging contrast modes, such as absorption and fluorescence, and the utilization of scanning focused beam or full-field imaging modes. The first published demonstration of aRCTIC introduced full-field operando spectrotomography, whereby a model copper-zeolite exhaust gas catalyst for selective catalytic reduction of $NO_x$ was imaged under a series of reaction conditions. This resulted in 3D spatially resolved XAS data at the Cu K edge, allowing the observation of chemical gradients in copper oxidation state and local coordination environment within a complex technical catalyst at work [43]. This also demonstrates possible application in both focused beam (scanning) mode and full-field acquisition mode. The current implementation of aRCTIC is at the P06 microprobe beamline of PETRA III, which can perform XAS tomography, XRF tomography, and XRD tomography, among other methods. Permutations of the aRCTIC setup have also been tested at beamline ID24 of the ESRF, and beamline microXAS of the Swiss Light Source [43], and beamline P64 of PETRA III. This shows that the aRCTIC concept may be readily adapted for various beamlines, even XAS or scattering beamlines.

The main advance of aRCTIC is that it enables operando tomography in various forms. Operando tomography is, in general, an underdeveloped technique in gas-phase heterogeneous catalysis and materials research, partly due to the deficit of appropriate sample environments. The aRCTIC concept aims to remedy this situation, providing a relatively simple and accessible means of performing operando tomography measurements of catalysts and functional materials, including quantitative performance data. During the development of the aRCTIC setup and based on earlier literature, a crucial compromise was uncovered between (i) having a capillary sample holder open at one end (i.e., not gas-tight),



which puts fewer constraints on translation or rotation up to 180 or 360°, but introduces severe limitations on the gas flow rate and composition both for safety reasons (e.g., toxic, flammable gases), and to prevent back-diffusion of ambient air reaching the sample [40,53]; (ii) having a closed gas-tight system with a structurally stable, but X-ray opaque support frame, thereby resulting in the acquisition of missing-wedge data (180°) and introducing artifacts during tomography reconstruction, possibly compromising resolution and quality of the images obtained [38,39,54]. Both such setups have been successfully applied in the literature for imaging catalysts at work. The aRCTIC setup achieves a compromise between these previous studies while providing a crucial advantage. Using aRCTIC, a controlled gas environment is maintained (as in case (ii)) while acquiring tomography data over the full 180° angular range (as in case (i)), without introducing significant artifacts from the support rod. This is achieved through the use of glassy carbon, which is reasonably transparent to hard X-rays (furthermore tested at Cu K edge [43]) while being rigid enough to maintain the sample at a fixed position and resist torsion or breakage during sample rotation. This makes aRCTIC the only known setup at present, which can deliver and maintain a precisely controlled gas and temperature environment, with quantitative analysis of reaction products during heterogeneous gas-phase catalysis and unobstructed 180° tomographic measurements [43].

Despite the capabilities of aRCTIC, the concept exhibits some limitations currently. The current iteration of aRCTIC has not been verified above ambient pressures, although in principle, gas delivery lines composed of polymer (e.g., PFA, PTFE) can be selected to allow elevated pressure. In addition, only gas-phase catalysis experiments have been performed currently, although the setup has the potential for use in liquid phase flow chemistry experiments due to the contained sample environment. A more significant challenge comes from the complex procedure of tomography measurements which require rotation and translation over many sampling points. This results in typical scan times on the order of hours, so that time-resolved measurements are not currently feasible, at least with the sample size and target resolution chosen here. While using smaller samples or inferior spatial resolution is certainly possible and would greatly accelerate the scan procedure, in catalysis research, this can reduce either the representativeness or accuracy of the measurement, respectively. A certain reduction in scan time can be achieved by optimizing scanning protocols, e.g., continuous 360° rotation or simultaneous rotation and translation, but to a certain extent, the hard limit on scan time comes from the source flux, which is independent of the sample environment used here. In the current experiment, a photon flux on the order of $10^9$ ph/s was present (during XRD tomography studies), but the acquisition time per scanning point was set at 300 ms to compensate for beam attenuation. Assuming the requirement to measure a sample of the same thickness, the bottleneck in the current measurements was, therefore, limited by the source flux. Improvement of the source flux can greatly accelerate the acquisition process irrespective of the acquisition mode selected, e.g., XAS, fluorescence, diffraction. Nevertheless, the aRCTIC concept is highly robust and enables relatively straightforward acquisition of operando tomography data. Further experimental possibilities and optimization of scan times will be actively developed going forward.

Operando hard X-ray tomography has considerable potential in catalysis research due to the range of information which can be obtained, and the spatially resolved manner with which it is obtained. For example, measurement of chemical or physical gradients in both lateral and radial directions within catalytic reactors [43] and the elucidation of dilute or minority species, which are in principle below the detection limit of conventional XAS or XRD [40,54]. With the arrival of extremely brilliant, near-diffraction-limited synchrotron light sources in the coming years, it is clear that X-ray microscopy will gain considerable advantages, particularly in terms of spatial resolution, measurement time, and chemical sensitivity. The need for appropriate sample environments and methodologies to take advantage of these new experimental capabilities is an important concern, which is addressed here.

## 3. Materials and Methods

### 3.1. Instrumentation of aRCTIC

The following axis definitions are used here vertical (z-axis), horizontal plane (x, y-axes). The aRCTIC design is flexible and may be adapted for different beamlines. The basic setup was designed around a PRS-110 precision rotation stage (PI miCos, Eschbach, Germany) mounted horizontally for sample rotation around the z-axis. A pair of miniature translation stages (OWIS, Staufen im Breisgau, Germany) mounted on top of the rotation stage allows the manual center of rotation alignment (x, y-axes) of the sample concerning the rotation stage. A tilting base plate fixed into the upper translation stage and supported by three springs acts as a goniometer for manual tilt angle correction (x–z or y–z plane) and vertical alignment of the sample holder. These components are generally interchangeable with alternative motor systems depending on the specific beamline and do not play a particular role in the in situ/operando functionality of the aRCTIC sample environment.

### 3.2. Implementation of aRCTIC at P06 Microprobe Beamline

The implementation of aRCTIC at P06 used an alternative motor setup to that indicated above. At the base was a UPR-270 rotation stage (PI miCos, Eschbach, Germany), with a pair of 5101.07 stages (HUBER Diffraktionstechnik GmbH & Co. KG, Rimsting, Germany) for sample rotation and center of rotation alignment, respectively. The tilting base plate was used as a manual goniometer as described above. During tomography scans, the entire motor stack described above was translated horizontally and vertically using NPE-200 and LMS-230 stages (PI miCos, Eschbach, Germany), respectively. The inlet gas tube was attached to the tilting base plate by means of a stainless-steel bulkhead fitting (Swagelok, Solon, OH, USA). The tilting base plate also contains a clamp fitting for the glassy carbon support rod (HTW Hochtemperatur-Werkstoffe, Thierhaupten, Germany), which was positioned vertically. A top plate containing a second clamp and bulkhead fitting allows fixed positioning of the support rod and connection of the outlet gas tube, respectively. The upper gas tube was positioned vertically and loosely clamped at ca. 30 cm directly above the top plate to avoid collision with other beamline components during rotation and allowing sufficient slack to perform rotation without significant torsion. The sample itself was placed in a quartz capillary (compatible sizes 500 μm to 3 mm tested) fixed between the two bulkhead fittings (Swagelok, Solon, OH, USA) with epoxy glue, producing a gas-tight system. The glassy carbon support bore the weight of the top plate and prevented torsional force on the fragile capillary, allowing stable rotation of the capillary containing the sample. In the current study, a pair of hot air blowers were positioned with standard laboratory clamps according to the sample height to provide bidirectionally uniform heating.

### 3.3. Catalytic Tests at P06 Microprobe Beamline

The aRCTIC setup was installed at the microprobe end station of beamline P06 of PETRA III (DESY, Hamburg, Germany). Approximately 2.4 mg of 20 wt % $Ni/Al_2O_3$ (dried gel after synthesis) [45] was loaded into a quartz capillary of 1 mm outer diameter (particle size 100–200 μm, bed length ca. 6 mm) and held in position using quartz wool plugs. The capillary was fixed in the aRCTIC setup with thermally resistant epoxy glue. Gas mixtures were delivered using mass flow controllers (Bronkhorst, Ruurlo, The Netherlands). Temperature conditions were applied by twin hot air blowers (LE mini sensor, Leister Technologies, Kägiswil, Switzerland), each connected to individual Eurotherm 2416 temperature controllers (Schneider Electric, Rueil-Malmaison, France), which were controlled by a Labview software (National Instruments, Austin, TX, USA). The following experimental conditions were applied in sequence: (i) ex situ—pure He, 10 mL/min, ambient temperature; (ii) reducing conditions—25/75% $H_2$/He, 10 mL/min, 823 K; (iii) reaction conditions—20/5/75% $H_2$/$CO_2$/He, 10 mL/min, 673 K; (iv) thermal aging in reaction conditions, 973 K, and (v) reaction conditions after aging, 673 K. The difference between the programmed temperature of the gas blowers and the temperature experienced by the catalysts was resolved by initially performing a manual calibration from 323 to 973 K

with a portable type K thermocouple, which was held close to the sample position (see Supplementary Materials). The temperature was consistently checked at several intervals throughout the measurements. The gas outlet of the reactor was analyzed by an Omnistar 320 MS (Pfeiffer Vacuum, Germany) connected downstream. The following mass traces were monitored: $H_2$, He, $CH_4$, $H_2O$, CO, $O_2$, Ar, $CO_2$ (m/z = 2, 4, 15, 16, 18, 28, 32, 40, 44).

### 3.4. XRD-CT Data Acquisition

XRD-CT experiments were performed at the microprobe end station of beamline P06 of PETRA III (DESY, Hamburg, Germany). XRD tomography measurements were performed on individual slices of the sample using a scanning pencil beam of $4 \times 4$ $\mu m^2$ at 35 keV ($\lambda$ = 0.0354 nm), formed using a stack of compound refractive lenses. The energy was chosen to limit attenuation by the sample, and the beam size was selected to ensure powder diffraction conditions during tomography scans. The photon flux was on the order of some $10^9$ ph/s. Tomography data were acquired on individual slices using a translate-rotate measurement scheme. At each beam position, a full XRD pattern was recorded in a 2$\theta$ range from 0 to 22° using an Eiger4M detector approximately 30 cm downstream of the sample. The 1 mm wide capillary with the sample was translated continuously over 1400 $\mu m$, with data acquired at 10 $\mu m$ intervals and 300 ms acquisition time per point, resulting in some air buffer on both sides of the sample. For tomography acquisition, a total of 181° angular range was measured in 1° steps, resulting in 181 projections. The XRD measurements were calibrated using a $LaB_6$ standard.

### 3.5. XRD-CT Data Treatment

The principles of XRD tomography data treatment can be found in the literature and are briefly summarized here [40]. The obtained 2D diffraction patterns from the Eiger4M detector were transformed to 1D XRD patterns by azimuthal integration using the pyFAI library [55]. Each of the diffraction patterns was binned into 999 individual bins representing 2$\theta$ values. The data recorded over all scan points for each 2$\theta$ value was treated as an individual tomogram in the reconstruction process, resulting in 999 tomograms (or 2$\theta$ values) for each completed scan. The measurement scheme using the new setup leads to some special considerations for preprocessing the data. Especially, the support rod of the aRCTIC setup eclipses the sample over a 5–10° angular range, producing a relatively stronger attenuation or diffraction signal, which may need to be compensated or mitigated before reconstruction (see Supplementary Materials). Tomographic reconstruction was performed using the simultaneous iterative reconstruction technique (SIRT, 100 iterations) algorithm as implemented in the ASTRA Toolbox [56]. The spatial resolution of the reconstructed tomograms was estimated as 20 $\mu m$ by plotting the signal intensity along a sharp edge feature of the sample (see Supplementary Materials).

### 3.6. Catalytic Tests at ESRF ID24 ED-XAS Beamline

The aRCTIC setup was installed at the ED-XAS beamline ID24 of ESRF (Grenoble, France). Approximately 1 mg of 2 wt % $Pt/Al_2O_3$ (precalcined in air at 773 K) was loaded into a quartz capillary of 1 mm outer diameter (particle size 100–200 $\mu m$, bed length ca. 3 mm), which was placed within the aRCTIC setup. The same gas and temperature delivery system, temperature calibration procedure, and product analysis by MS were used as described above. The following experimental conditions were applied in sequence: (i) inert conditions—2 mL/min He at 293 K; (ii) 30 mL/min $CH_4/O_2$/He in 3/1.5/95.5% at 553 K, followed by (iii) at 673 K. The following mass traces were monitored by MS at the reactor outlet: $H_2$, He, $CH_4$, $H_2O$, CO, $O_2$, Ar, $CO_2$, (m/z = 2, 4, 15, 18, 28, 32, 40, 44).

### 3.7. ED-XAS-CT Data Acquisition

ED-XAS-CT measurements were performed at beamline ID24 of the European Synchrotron Radiation Facility (Grenoble, France) [57]. ED-XAS-CT measurements were performed using a polychromatic beam. The beam was generated with a cross-section

of $3 \times 7$ µm$^2$ (horizontal $\times$ vertical) by focusing horizontally with a bent Si (111) polychromator and vertically using a bent Si mirror. This resulted in a fan-shaped beam with position-sensitive energy distribution. The position-sensitive detector used 1001 pixels to resolve an energy range of around 11,370 to 11,970 eV simultaneously. The photon flux was on the order of some 10$^{13}$ ph/s. Tomography data were acquired on individual slices using a translate-rotate measurement scheme. The 1 mm wide capillary with the sample was translated in steps over 1700 µm, with data acquired at 4 µm intervals and 100 ms acquisition time per point. For tomography acquisition, a total of 180° angular range was measured in 0.7° steps, resulting in 256 projections.

### 3.8. ED-XAS-CT Data Treatment

The principles of tomo-XANES can be found in the literature and are briefly summarized here [50]. Due to the ED-XAS detection mode, each sampling point contained 1001 individual measurements, one per detector channel. By using an energy-discriminating detector, each pixel was effectively recorded 1001 times in a single 100 ms interval across an energy range from 11,370 to 11,970 keV. By translating the sample perpendicular to the beam axis, each horizontal slice (equivalent to one projection) was, therefore, measured 1001 times simultaneously. Repeating this process at different rotational angles, therefore, resulted in the acquisition of 1001 distinct sinograms simultaneously. After reconstruction, this resulted in 1001 individual tomograms from a single measurement, one per energy. Tomographic reconstruction was performed using the simultaneous iterative reconstruction technique (SIRT, 100 iterations) algorithm as implemented in the ASTRA Toolbox [56]. An overview of the reconstruction steps is shown in the Supplementary Materials. The incident X-ray energy was measured by moving the reactor out of the beam, providing a direct path between the polychromator and detector.

### 4. Conclusions

The aRCTIC setup enables operando hard X-ray tomography for 3D spatially resolved structural analysis of heterogeneous catalysts at work. The setup is compatible with a range of X-ray imaging contrast modes (e.g., XAS, XRD, and others) and supports both scanning-focused beam and full-field imaging modes. Precise control of gas and temperature conditions allows for meaningful quantitative gas-phase activity tests combined with tomography. The flexible operating conditions of aRCTIC are suitable for probing numerous gas-phase catalytic reactions. Operando tomography is highlighted as a developing range of techniques for probing structure–activity relations in complex heterogeneous catalytic systems and is relevant in energy materials research. The development of highly brilliant, diffraction-limited synchrotron light sources is proposed to greatly improve spatial resolution, scan times and photon flux, particularly for X-ray microscopy and tomography. The aRCTIC setup is introduced as a flexible future-ready concept, which can exploit established and developing tomographic imaging methods to develop meaningful catalyst structure–activity relations with 3D spatial resolution.

**Supplementary Materials:** The following are available online at https://www.mdpi.com/article/10.3390/catal11040459/s1, Figure S1: Mass spectrometry data—$CO_2$ methanation studies, Figure S2: Calculation of $CO_2$ conversion, Figure S3: Hot air blower calibration curve, Figure S4: Schematic of the aRCTIC setup, Figure S5: Removal of support rod from sinograms—$CO_2$ methanation studies, Figure S6: Summary of tomography reconstruction steps—methane CPO studies, Figure S7: Sharp edge fitting, Figure S8: Overview of ED-XAS tomography data measured at 673 K, Figure S9: Overview of MS signals from ED-XAS-CT at 523 K, Figure S10: Overview of MS signals from ED-XAS-CT at 673 K.

**Author Contributions:** Conceptualization, D.F.S., D.M.M., J.-D.G. and T.L.S.; methodology, J.B., D.F.S., J.G., G.F., D.M.M., S.P., T.L.S.; software, D.F.S., J.G., G.F., D.M.M., S.P.; validation, J.B., S.W., D.F.S., D.E.D., T.L.S.; formal analysis, J.B., S.W., D.F.S., D.E.D., T.L.S.; investigation, J.B., S.W., D.E.D., J.G., D.M.M., T.L.S.; data curation, J.B., S.W.; writing—original draft preparation, J.B., S.W., D.E.D., T.L.S.; writing—review and editing, all co-authors; visualization, J.B., S.W., D.F.S., J.G., D.M.M., T.L.S.;

supervision, J.-D.G. and T.L.S.; project administration, J.-D.G. and T.L.S.; funding acquisition, J.-D.G. and T.L.S. All authors have read and agreed to the published version of the manuscript.

**Funding:** This work was supported by the German Federal Ministry of Education and Research (BMBF) projects "MicTomoCat" (05K16VK1) and "COSMIC" (05K19VK4). Gefördert durch die Deutsche Forschungsgemeinschaft (DFG)-406914011, funded by the Deutsche Forschungsgemeinschaft (German Research Foundation)-406914011.

**Data Availability Statement:** X-ray imaging data supporting the study are available from the authors upon reasonable request. The data are not available publicly due to large file sizes (TB scale).

**Acknowledgments:** We acknowledge DESY (Hamburg, Germany), a member of the Helmholtz Association HGF, for the provision of experimental facilities. Operando XRD tomography experiments were carried out on beamline P06 of PETRA III, and we would like to thank Michael Stuckelberger, Srashtasrita Das, Linda Klag and Mariam Schulte for their assistance during the measurements. Beamtime was allocated for proposal I-20191105. We acknowledge Ken Luca Abel and Roger Gläser (both Leipzig University) for providing the Ni/Al$_2$O$_3$ catalyst sample. Additional tests during the development of the aRCTIC setup were performed at beamline X05LA (microXAS) of the Paul Scherrer Institut (Villigen, Switzerland), beamline ID24 of the European Synchrotron Radiation Facility (ESRF, Grenoble, France), and the CAT-ACT beamline of the KIT Synchrotron (Karlsruhe, Germany). We thank the Institute for Beam Physics and Technology (IBPT) at Karlsruhe Institute of Technology (KIT) for the operation of the KIT Synchrotron. We acknowledge Anna Zimina for discussions and tests at CAT-ACT. Darma Yuda is acknowledged for discussions and assistance in designing the aRCTIC setup. We acknowledge support by the KIT-Publication Fund of the Karlsruhe Institute of Technology.

**Conflicts of Interest:** The authors declare no conflict of interest. The funders had no role in the study's design, in the collection, analyses, or interpretation of data, in the writing of the manuscript, or in the decision to publish the results.

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
