# Peer review of "Sample Environment for Operando Hard X-ray Tomography—An Enabling Technology for Multimodal Characterization in Heterogeneous Catalysis"

_catalysts, doi:10.3390/catal11040459_

Round 1
Reviewer 1 Report
It was a real pleasure to review the manuscript by Becher and co-authors. The authors report on the developped setup for operando X-ray tomography synchrotron experiments, which was successfully tested at two experimental beamlines. This is a new and very promising field for catalytic research. In my opinion, the authors make a good contribution to the field of catalysis and is definitely suited for Catalysts journal. The manuscript is already well written and can be accepted in its present form.
Reviewer 2 Report
This paper presents a flexible sample environment for operando hard X-ray tomography at synchrotron radiation sources. Examples of CO2 methanation and methane partial oxidation are shown as case study. The structure of this manuscript flows very well and the scientific aspect of this paper is significant.
Some minor issues in this paper
Page 5: legend of Figure 2: 15, CH3. CH3+ is an indication of CH4 formation. Therefore, it is better to change it to 15, CH4.
Figure S3: linear fit of the equation has both + and – symbol, which needs to be corrected.
